# A multi-agent control framework for co-adaptation in brain-computer interfaces

*Josh Merel[1], *Roy Fox[2], Tony Jebara[3], Liam Paninski[4]
[1]Department of Neurobiology and Behavior, [3]Department of Computer Science,
[4]Department of Statistics, Columbia University, New York, NY 10027
[2]School of Computer Science and Engineering, Hebrew University, Jerusalem 91904, Israel
jsm2183@columbia.edu, royf@cs.huji.ac.il,
jebara@cs.columbia.edu, liam@stat.columbia.edu

## Abstract

In a closed-loop brain-computer interface (BCI), adaptive decoders are used to learn parameters suited to decoding the user's neural response. Feedback to the user provides information which permits the neural tuning to also adapt. We present an approach to model this process of co-adaptation between the encoding model of the neural signal and the decoding algorithm as a multi-agent formulation of the linear quadratic Gaussian (LQG) control problem. In simulation we characterize how decoding performance improves as the neural encoding and adaptive decoder optimize, qualitatively resembling experimentally demonstrated closed-loop improvement. We then propose a novel, modified decoder update rule which is aware of the fact that the encoder is also changing and show it can improve simulated co-adaptation dynamics. Our modeling approach offers promise for gaining insights into co-adaptation as well as improving user learning of BCI control in practical settings.

## 1   Introduction

Neural signals from electrodes implanted in cortex [1], electrocorticography (ECoG) [2], and electroencephalography (EEG) [3] all have been used to decode motor intentions and control motor prostheses. Standard approaches involve using statistical models to decode neural activity to control some actuator (e.g. a cursor on a screen [4], a robotic manipulator [5], or a virtual manipulator [6]). Performance of offline decoders is typically different from the performance of online, closed-loop decoders where the user gets immediate feedback and neural tuning changes are known to occur [7, 8]. In order to understand how decoding will be performed in closed-loop, it is necessary to model how the decoding algorithm updates and neural encoding updates interact in a coordinated learning process, termed co-adaptation.

There have been a number of recent efforts to learn improved adaptive decoders specifically tailored for the closed loop setting [9, 10], including an approach relying on stochastic optimal control theory [11]. In other contexts, emphasis has been placed on training users to improve closed-loop control [12]. Some efforts towards modeling the co-adaptation process have sought to model properties of different decoders when used in closed-loop [13, 14, 15], with emphasis on ensuring the stability of the decoder and tuning the adaptation rate. One recent simulation study also demonstrated how modulating task difficulty can improve the rate of co-adaptation when feedback noise limits performance [16]. However, despite speculation that exploiting co-adaptation will be integral to state-of-the-art BCI [17], general models of co-adaptation and methods which exploit those models to improve co-adaptation dynamics are lacking.

---

We propose that we should be able to leverage our knowledge of how the encoder changes in order to better update the decoder. In the current work, we present a simple model of the closed-loop co-adaptation process and show how we can use this model to improve decoder learning on simulated experiments. Our model is a novel control setting which uses a split Linear Quadratic Gaussian (LQG) system. Optimal decoding is performed by Linear Quadratic Estimation (LQE), effectively the Kalman filter model. Encoding model updates are performed by the Linear Quadratic Regulator (LQR), the dual control problem of the Kalman filter. The system is split insofar as each agent has different information available and each performs optimal updates given the state of the other side of the system. We take advantage of this model from the decoder side by anticipating changes in the encoder and pre-emptively updating the decoder to match the estimate of the further optimized encoding model. We demonstrate that this approach can improve the co-adaptation process.

## 2 Model framework

### 2.1 Task model

For concreteness, we consider a motor-cortical neuroprosthesis setting. We assume a naive user, placed into a BCI control setting, and propose a training scheme which permits the user and decoder to adapt. We provide a visual target cue at a 3D location and the user controls the BCI via neural signals which, in a natural setting, relate to hand kinematics. The target position is moved each timestep to form a trajectory through the 3D space reachable by the user's hand. The BCI user receives visual feedback via the displayed location of their decoded hand position. The user's objective is to control their cursor to be as close to the continuously moving target cursor as possible. A key feature of this scheme is that we know the "intention" of the user, assuming it corresponds to the target.

The complete graphical model of this system is provided in figure 1. $x_t$ in our simulations is a three dimensional position vector (Cartesian Coordinates) corresponding to the intended hand position. This variable could be replaced or augmented by other variables of interest (e.g. velocity). We randomly evolve the target signal using a linear-Gaussian drift model (eq. (1)). The neural encoding model is linear-Gaussian in response to intended position $x_t$ and feedback $\hat{x}_{t-1}$ (eq. (2)), giving a vector of neural responses $u_t$ (e.g. local field potential or smoothed firing rates of neural units). Since we do not observe the whole brain region, we must subsample the number of neural units from which we collect information. The transformation $C$ is conceptually equivalent to electrode sampling and $y_t$ is the observable neural response vector via the electrodes (eq. (3)). Lastly, $\hat{x}_t$ is the decoded hand position estimate, which also serves as visual feedback (eq. (4)).

$$x_t = Px_{t-1} + \xi_t; \qquad\qquad \xi_t \sim \mathcal{N}(0, Q) \qquad (1)$$

$$u_t = Ax_t + B\hat{x}_{t-1} + \eta_t; \qquad\qquad \eta_t \sim \mathcal{N}(0, R) \qquad (2)$$

$$y_t = Cu_t + \epsilon_t; \qquad\qquad \epsilon_t \sim \mathcal{N}(0, S) \qquad (3)$$

$$\hat{x}_t = Fy_t + G\hat{x}_{t-1}. \qquad\qquad (4)$$

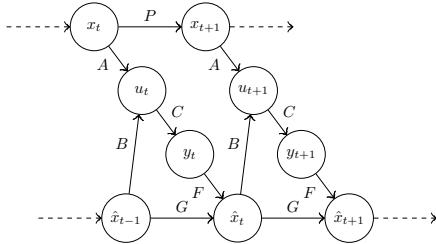

Figure 1: Graphical model relating target signal ($x_t$), neural response ($u_t$), electrode observation of neural response ($y_t$), and decoded feedback signal ($\hat{x}_t$).

During training, the decoding system is allowed access to the target position, interpreted as the real intention $x_t$. The decoded $\hat{x}_t$ is only used as feedback, to inform the user of the gradually learned dynamics of the decoder. After training, the system is tested on a task with the same parameters of the trajectory dynamics, but with the actual intention only known to the user, and hidden from the decoder. A natural objective is to minimize tracking error, measured as accumulated mean squared error between the target and neurally decoded pose over time.

For contemporary BCI applications, the Kalman filter is a reasonable baseline decoder, so we do not consider even simpler models. However, for other applications one might wish to consider a model in which the state at each timestep is encoded independently. It is possible to find a closed form for the optimal encoder and decoder that minimizes the error in this case [18, 19].

Sections 2.2 and 2.3 describe the model presented in figure 1 as seen from the distinct viewpoints of the two agents involved – the encoder and the decoder. The encoder observes $x_t$ and $\hat{x}_{t-1}$, and selects $A$ and $B$ to generate a control signal $u_t$. The decoder observes $y_t$, and selects $F$ and $G$ to estimate the intention as $\hat{x}_t$. We assume that both agents are free to performed unconstrained optimization on their parameters.

## 2.2 Encoding model and optimal decoder

Our encoding model is quite simple, with neural units responding in a linear-Gaussian fashion to intended position $x_t$ and feedback $\hat{x}_{t-1}$ (eq. (2)). This is a standard model of neural responses for BCI. The matrices $A$ and $B$ effectively correspond to the tuning response functions of the neural units, and we will allow these parameters to be adjusted under the control of the user. The matrix $C$ corresponds to the observation of the neural units by the electrodes, so we treat it as fixed (in our case $C$ will down-sample the neurons). For this paper, we assume noise covariances are fixed and known, but this can be generalized. Given the encoder, the decoder will estimate the intention $x_t$, which follows a hidden Markov chain (eq. (1)). The observations available to the decoder are the electrode samples $y_t$ (eq. (2) and (3))

$$y_t = CAx_t + CB\hat{x}_{t-1} + \epsilon'_t; \qquad\qquad \epsilon'_t \sim \mathcal{N}(0, R_C) \qquad\qquad (5)$$
$$R_C = CRC^T + S. \qquad\qquad\qquad\qquad\qquad\qquad\qquad\qquad (6)$$

Given all the electrode samples up to time $t$, the problem of finding the most likely hidden intention is a Linear-Quadratic Estimation problem (figure 2), and its standard solution is the Kalman filter, and this decoder is widely in similar contexts. To choose appropriate Kalman gain $F$ and mean dynamics $G$, the decoding system needs a good model of the dynamics of the underlying intention process ($P$, $Q$ of eq.(1)) and the electrode observations ($CA$, $CB$, and $R_C$ of eqs. (5) & (6)). We can assume that $P$ and $Q$ are known since the decoding algorithm is controlled by the same experimenter who specifies the intention process for the training phase. We discuss the estimation of the observation model in section 4.

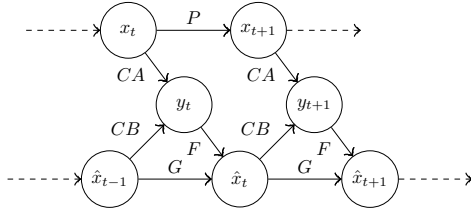 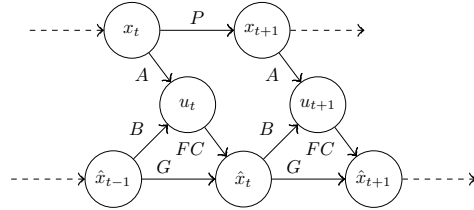

Figure 2: Decoder's point of view – target signal ($x_t$) directly generates observed responses ($y_t$), with the encoding model collapsed to omit the full signal ($u_t$). Decoded feedback signal ($\hat{x}_t$) is generated by the steady state Kalman filter.

Figure 3: Encoder's point of view – target signal ($x_t$) and decoded feedback signal ($\hat{x}_{t-1}$) generate neural response ($u_t$). Model of decoder collapses over responses ($y_t$) which are unseen by the encoder side.

Given an encoding model, and assuming a very long horizon [1], there exist standard methods to optimize the stationary value of the decoder parameters [20]. The stationary covariance $\Sigma$ of $x_t$ given $\hat{x}_{t-1}$ is the unique positive-definite fixed point of the Riccati equation

$$\Sigma = P\Sigma P^T - P\Sigma(CA)^T(R_C + (CA)\Sigma(CA)^T)^{-1}(CA)\Sigma P^T + Q. \qquad (7)$$

The Kalman gain is then

$$F = \Sigma(CA)^T((CA)\Sigma(CA)^T + R_C)^{-1} \qquad\qquad (8)$$

with mean dynamics

$$G = P - F(CA)P - F(CB). \qquad\qquad (9)$$

We estimate $\hat{x}_t$ using eq. (4), and this is the most likely value, as well as the expected value, of $x_t$ given the electrode observations $y_1, \ldots, y_t$. Using this estimate as the decoded intention is equivalent to minimizing the expectation of a quadratic cost

$$c_{lqe} = \sum_t \tfrac{1}{2}\|x_t - \hat{x}_t\|^2. \tag{10}$$

## 2.3 Model of co-adaptation

At the same time as the decoder-side agent optimizes the decoder parameters $F$ and $G$, the encoder-side agent can optimize the encoder parameters $A$ and $B$. We formulate encoder updates for the BCI application as a standard LQR problem. This framework requires that the encoder-side agent has an intention model (same as eq. (1)) and a model of the decoder. The decoder model combines eq. (3) and (4) into

$$\hat{x}_t = FCu_t + G\hat{x}_{t-1} + F\epsilon_t. \tag{11}$$

This model is depicted in figure 3. We assume that the encoder has access to a perfect estimate of the intention-model parameters $P$ and $Q$ (task knowledge). We also assume that the encoder is free to change its parameters $A$ and $B$ arbitrarily given the decoder-side parameters (which it can estimate as discussed in section 4).

As a model of real neural activity, there must be some cost to increasing the power of the neural signal. Without such a cost, the solutions diverge. We add an additional cost term (a regularizer), which is quadratic in the magnitude of the neural response $u_t$, and penalizes a large neural signal

$$c_{lqr} = \sum_t \tfrac{1}{2}\|x_t - \hat{x}_t\|^2 + \tfrac{1}{2}u_t^T \tilde{R} u_t. \tag{12}$$

Since the decoder has no direct influence on this additional term, it can be viewed as optimizing for this target cost function as well. The LQR problem is solved similarly to eq. (7), by assuming a very long horizon and optimizing the stationary value of the encoder parameters [20].

We next formulate our objective function in terms of standard LQR parameters. The control depends on the joint process of the intention and the feedback $(x_t, \hat{x}_{t-1})$, but the cost is defined between $x_t$ and $\hat{x}_t$. To compute the expected cost given $x_t$, $\hat{x}_{t-1}$ and $u_t$, we use eq. (11) to get

$$\mathbf{E}\,\|\hat{x}_t - x_t\|^2 = \|FCu_t + G\hat{x}_{t-1} - x_t\|^2 + \text{const} \tag{13}$$
$$= (G\hat{x}_{t-1} - x_t)^T(G\hat{x}_{t-1} - x_t) + (FCu_t)^T(FCu_t) + 2(G\hat{x}_{t-1} - x_t)^T(FCu_t) + \text{const}.$$

Equation 13 provides the error portion of the quadratic objective of the LQR problem. The standard solution for the stationary case involves computing the Hessian $V$ of the cost-to-go in joint state $\begin{bmatrix} x_t \\ \hat{x}_{t-1} \end{bmatrix}$ as the unique positive-definite fixed point of the Riccati equation

$$V = \tilde{P}^T V \tilde{P} + (\tilde{N} + \tilde{P}^T V \tilde{D})(\tilde{R} + \tilde{S} + \tilde{D}^T V \tilde{D})^{-1}(\tilde{N}^T + \tilde{D}^T V \tilde{P}) + \tilde{Q}. \tag{14}$$

Here $\tilde{P}$ is the process dynamics for the joint state of $x_t$ and $\hat{x}_{t-1}$ and $\tilde{D}$ is the controllability of this dynamics. $\tilde{Q}$, $\tilde{S}$ and $\tilde{N}$ are the cost parameters which can be determined by inspection of eq. (13). $\tilde{R}$ is the Hessian of the neural response cost term which is chosen in simulations so that the resulting increase in neural signal strength is reasonable.

$$\tilde{P} = \begin{bmatrix} P & 0 \\ 0 & G \end{bmatrix}, \quad \tilde{D} = \begin{bmatrix} 0 \\ FC \end{bmatrix}, \quad \tilde{Q} = \begin{bmatrix} I & -G^T \\ -G & G^T G \end{bmatrix}, \quad \tilde{S} = (FC)^T(FC), \quad \tilde{N} = \begin{bmatrix} -FC \\ G^T(FC) \end{bmatrix}.$$

In our formulation, the encoding model $(A, B)$ is equivalent to the feedback gain

$$[A \quad B] = -(\tilde{D}^T V \tilde{D} + \tilde{R} + \tilde{S})^{-1}(\tilde{N}^T + \tilde{D}^T V \tilde{P}). \tag{15}$$

This is the optimal stationary control, and is generally not optimal for shorter planning horizons. In the co-adaptation setting, the encoding model $(A_t, B_t)$ regularly changes to adapt to the changing decoder. This means that $(A_t, B_t)$ is only used for one timestep (or a few) before it is updated. The effective planning horizon is thus shortened from its ideal infinity, and now depends on the rate and magnitude of the perturbations introduced in the encoding model. Eq. (14) can be solved for this finite horizon, but here for simplicity we assume the encoder updates introduce small or infrequent enough changes to keep the planning horizon very long, and the stationary control close to optimal.

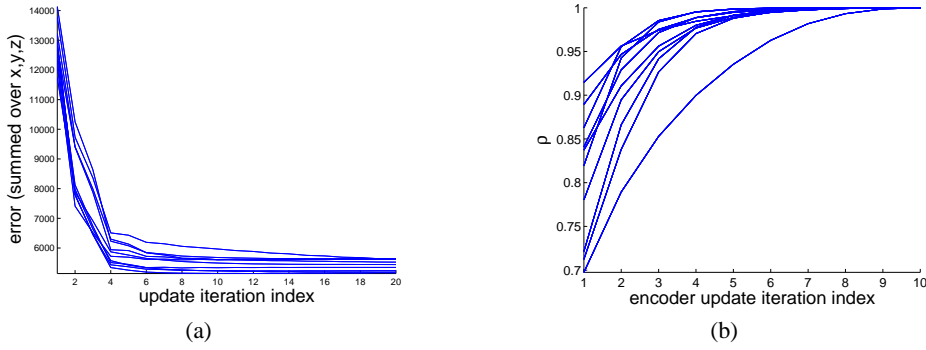

(a)                                      (b)

Figure 4: (a) Each curve plots single trial changes in decoding mean squared error (MSE) over whole timeseries as a function of the number of update half-iterations. The encoder is updated in even steps, the decoder in odd ones. Distinct curves are for multiple, random initializations of the encoder. (b) Plots the corresponding changes in encoder parameter updates - y-axis, $\rho$, is correlation between the vectorized encoder parameters after each update with the final values.

## 3    Perfect estimation setting

We can consider co-adaptation in a hypothetical setting where each agent has instant access to a perfect estimate of the other's parameters as soon as they change. To keep this setting comparable to the setting of section 4, where parameter estimation is needed, we only allow each agent access to those variables that it could, in principle, estimate. We assume both agents know the parameters $P$ and $Q$ of the intention dynamics, that the encoder knows $FC$ and $G$ of eq. (11), and that the decoder knows $CA$, $CB$ and $R_C$ of eq. (5) and (6). These are the same parameters needed by each agent for its own re-optimization. This process of parameter updates is performed by alternating between the encoder update equations (7)-(9) and the decoder update equations (14)-(15). Since the agents take turns minimizing the expected infinite-horizon objectives of eq. (12) given the other, this cost will tend to decrease, approximately converging.

Note that neither of these steps depends explicitly on the observed values of the neural signal $u_t$ or the decoded output $\hat{x}_t$. In other words, co-adaptation can be simulated without ever actually generating the stochastic process of intention, encoding and decoding. However, this process and the signal-feedback loop become crucial when estimation is involved, as in section 4. Then each agent's update indirectly depends on its observations through its estimated model of the other agent.

To examine the dynamics in this idealized setting, we hold fixed the target trajectory $x_{1...T}$ as well as the realization of the noise terms. We initialize the simulation with a random encoding model and observe empirically that, as the encoder and the decoder are updated alternatingly, the error rapidly reduces to a plateau. As the improvement saturates, the joint encoder-decoder pair approximates a locally optimal solution to the co-adaptation problem. Figure 4(a) plots the error as a function of the number of model update iterations – the different curves correspond to distinct, random initializations of the encoder parameters $A, B$ with everything else held fixed. We emphasize that for a fixed encoder, the first decoder update would yield the infinite-horizon optimal update if the encoder could not adapt, and the error can be interpreted relative to this initial optimal decoding (see supplementary fig1(a) for depiction of initial error and improvement by encoder adaptation in supplementary fig1(b)). This method obtains optimized encoder-decoder pairs with moderate sensitivity to the initial parameters of the encoding model. Interpreted in the context of BCI, this suggests that the initial tuning of the observed neurons may affect the local optima attainable for BCI performance due to standard co-adaptation. We may also be able to optimize the final error by cleverly choosing updates to decoder parameters in a fashion which shifts which optimum is reached. Figure 4(b) displays the corresponding approximate convergence of the encoder parameters - as the error decreases, the encoder parameters settle to a stable set (the actual final values across initializations vary).

Parameters free from the standpoint of the simulation are the neural noise covariance $R_C$ and the Hessian $\tilde{R}$ of the neural signal cost. We set these to reasonable values - the noise to a moderate

level and the cost sufficiently high as to prevent an exceedingly large neural signal which would swamp the noise and yield arbitrarily low error (see supplement). In an experimental setting, these parameters would be set by the physical system and they would need to be estimated beforehand.

# 4  Partially observable setting with estimation

More realistic than the model of co-adaptation where the decoder-side and encoder-side agents automatically know each other's parameters, is one where the rate of updating is limited by the partial knowledge each agent has about the other. In each timestep, each agent will update its estimate of the other agent's parameters, and then use the current estimates to re-optimize its own parameters. In this work we use a recursive least squares (RLS) which is presented in the supplement section 3 for this estimation. RLS has a forgetting factor $\lambda$ which regulates how quickly the routine expects the parameters it estimates to change. This co-adaptation process is detailed in procedure 1. We elect to use the same estimation routine for each agent and assume that the user performs ideal-observer style optimal estimation. In general, if more knowledge is available about how a real BCI user updates their estimates of the decoder parameters, such a model could easily be used. We could also explore in simulation how various suboptimal estimation models employed by the user affect co-adaptation.

As noted previously, we will assume the noise model is fixed and that the decoder side knows the neural signal noise covariance $R_C$ (eq. (6)). The encoder-side will use a scaled identity matrix as the estimate of the electrodes-decoder noise model. To jointly estimate the decoder parameters and the noise model, an EM-based scheme would be a natural approach (such estimation of the BCI user's internal model of the decoder has been treated explicitly in [21]).

---

**Procedure 1** standard co-adaptation

    **for** $t = 1$ to $lengthTraining$ **do**
        **Encoder-side**
            Get $x_t$ and $\hat{x}_{t-1}$
            Update encoder-side estimate of decoder $\widehat{FC}, \widehat{G}$ (RLS)
            Update optimal encoder $A, B$ using current decoder estimate $\widehat{FC}, \widehat{G}$ (LQR)
            Encode current intention using $A, B$ and send signal $y_t$
        **Decoder-side**
            Get $x_t$ and $y_t$
            Update decoder-side estimate of encoder $\widehat{CA}, \widehat{CB}$ (RLS)
            Update optimal decoder $F, G$ using current encoder estimate $\widehat{CA}, \widehat{CB}$ (LQE)
            Decode current signal using $F, G$ and display as feedback $\hat{x}_t$
    **end for**

---

Standard co-adaptation yields improvements in decoding performance over time as the encoder and decoder agents estimate each others' parameters and update based on those estimates. Appropriately, that model will improve the encoder-decoder pair over time, as in the blue curves of figure 5 below.

# 5  Encoder-aware decoder updates

In this section, we present an approach to model the encoder updates from the decoder side. We will use this to "take an extra step" towards optimizing the decoder for what the anticipated future encoder ought to look like.

In the most general case, the encoder can update $A_t$ and $B_t$ in an unconstrained fashion at each timestep $t$. From the decoder side, we do not know $C$ and therefore we cannot know $FC$, an estimate of which is needed by the user to update the encoder. However, the decoder sets $F$ and can predict updates to $[CA \quad CB]$ directly, instead of to $[A \quad B]$ as the actual encoder does (equation 15). We emphasize that this update is not actually how the user will update the encoder, rather it captures how the encoder ought to change the signals observed by the decoder (from the decoder's perspective).

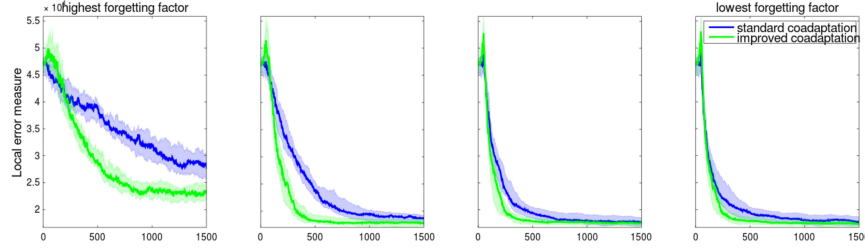

Figure 5: In each subplot, the blue line corresponds to decreasing error as a function of simulated time from standard co-adaptation (procedure 1). The green line corresponds to the improved one-step-ahead co-adaptation (procedure 2). Plots from left to right have decreasing RLS forgetting factor used by the encoder-side to estimate the decoder parameters. Curves depict the median error across 20 simulations with confidence intervals of 25% and 75% quantiles. Error at each timestep is appropriately cross-validated, it corresponds to taking the encoder-decoder pair of that timestep and computing error on "test" data.

We can find the update $[CA_{pred} \quad CB_{pred}]$ by solving a modified version of the LQR problem presented in section 2.3, eq. (15)

$$[CA_{pred} \quad CB_{pred}] = -(\tilde{D}'^T V \tilde{D}' + \tilde{R}' + \tilde{S}')^{-1}(\tilde{N}'^T + \tilde{D}'^T V \tilde{P}), \tag{16}$$

with terms defined similarly to section 2.3, except

$$\tilde{D}' = \begin{bmatrix} 0 \\ F \end{bmatrix}, \quad \tilde{S}' = F^T F, \quad \tilde{N}' = \begin{bmatrix} -F \\ G^T F \end{bmatrix}. \tag{17}$$

We also note that the quadratic penalty used in this approximation been transformed from a cost on the responses of all of the neural units to a cost only on the observed ones. $\tilde{R}'$ serves as a regularization parameter which now must be tuned so the decoder-side estimate of the encoding update is reasonable. For simplicity we let $\tilde{R}' = \gamma I$ for some constant coarsely tuned $\gamma$, though in general this cost need not be a scaled identity matrix. Equations 16 & 17 only use information available at the decoder side, with terms dependent on $FC$ having been replaced by terms dependent instead on $F$. These predictions will be used only to engineer decoder update schemes that can be used to improve co-adaptation (as in procedure 2).

---

**Procedure 2** r-step-ahead co-adaptation

    **for** $t = 1$ to $lengthTraining$ **do**
        **Encoder-side**
            As in section 5
        **Decoder-side**
            Get $x_t$ and $y_t$
            Update decoder-side estimate of encoder $\widehat{CA}, \widehat{CB}$ (RLS)
            Update optimal decoder $F, G$ using current encoder estimate $\widehat{CA}, \widehat{CB}$ (LQE)
            **for** $r = 1$ to $numStepsAhead$ **do**
                Anticipate encoder update $CA_{pred}, CB_{pred}$ to updated decoder $F, G$ (modified LQR)
                Update r-step-ahead optimal decoder $F, G$ using $CA_{pred}, CB_{pred}$ (LQE)
            **end for**
            Decode current signal using r-step-ahead $F, G$ and display as feedback $\widehat{x}_t$
    **end for**

---

The ability to compute decoder-side approximate encoder updates opens the opportunity to improve encoder-decoder update dynamics by anticipating encoder-side adaptation to guide the process towards faster convergence, and possibly to better solutions. For the current estimate of the encoder, we update the optimal decoder, anticipate the encoder update by the method of section above, and then update the decoder in response to the anticipated encoder update. This procedure allows r-step-ahead updating as presented in procedure 2. Figure 5 demonstrates how the one-step-ahead

scheme can improve the co-adaptation dynamics. It is not *a priori* obvious that this method would help - the decoder-side estimate of the encoder update is not identical to the actual update. An encoder-side agent more permissive of rapid changes in the decoder may better handle r-step-ahead co-adaptation. We have also tried r-step-ahead updates for $r > 1$. However, this did not outperform the one-step-ahead method, and in some cases yields a decline relative to standard co-adaptation. These simulations are susceptible to the setting of the forgetting factor used by each agent in the RLS estimation, the initial uncertainty of the parameters, and the quadratic cost used in the one-step-ahead approximation $\tilde{R}'$. The encoder-side RLS parameters in a real setting will be determined by the BCI user and $\tilde{R}'$ should be tuned (as a regularization parameter).

The encoder-side forgetting factor would correspond roughly to the plasticity of the BCI user with respect to the task. A high forgetting factor permits the user to tolerate very large changes in the decoder, and a low forgetting factor corresponds to the user assuming more decoder stability. From left to right in the subplots of figure 5, encoder-side forgetting factor decreases - the regime where augmenting co-adaptation may offer the most benefit corresponds to a user that is most uncertain about the decoder and willing to tolerate decoder changes. Whether or not co-adaptation gains are possible in our model depend upon parameters of the system. Nevertheless, for appropriately selected parameters, attempting to augment the co-adaptation should not hurt performance even if the user were outside of the regime where the most benefit is possible. A real user will likely perform their half of co-adaptation sub-optimally relative to our idealized BCI user and the structure of such suboptimalities will likely increase the opportunity for co-adaptation to be augmented. The timescale of these simulation results are unspecified, but would correspond to the timescale on which the biological neural encoding can change. This varies by task and implicated brain-region, ranging from a few training sessions [22, 23] to days [24].

## 6 Conclusion

Our work represents a step in the direction of exploiting co-adaptation to jointly optimize the neural encoding and the decoder parameters, rather than simply optimizing the decoder parameters without taking the encoder parameter adaptation into account. We model the process of co-adaptation that occurs in closed-loop BCI use between the user and decoding algorithm. Moreover, the results using our modified decoding update demonstrate a proof of concept that reliable improvement can be obtained relative to naive adaptive decoders by encoder-aware updates to the decoder in a simulated system. It is still open how well methods based on this approach will extend to experimental data.

BCI is a two-agent system, and we may view co-adaptation as we have formulated it within multi-agent control theory. As both agents adapt to reduce the error of the decoded intention given their respective estimates of the other agent, a fixed point of this co-adaptation process is a Nash equilibrium. This equilibrium is only known to be unique in the case where the intention at each timestep is independent [25]. In our more general setting, there may be more than one encoder-decoder pair for which each is optimal given the other. Moreover, there may exist non-linear encoders with which non-linear decoders can be in equilibrium. These connections will be explored in future work.

Obviously our model of the neural encoding and the process by which the neural encoding model is updated are idealizations. Future experimental work will determine how well our co-adaptive model can be applied to the real neuroprosthetic context. For rapid, low-cost experiments it might be best to begin with a human, closed-loop experiments intended to simulate a BCI [26]. As the Kalman filter is a standard decoder, it will be useful to begin experimental investigations with this choice (as analyzed in this work). More complicated decoding schemes also appear to improve decoding performance [23] by better accounting for the non-linearities in the real neural encoding, and such methods scale to BCI contexts with many output degrees of freedom [27]. An important extension of the co-adaptation model presented in this work is to non-linear encoding and decoding schemes. Even in more complicated, realistic settings, we hope the framework presented here will offer similar practical benefits for improving BCI control.

**Acknowledgments**

This project is supported in part by the Gatsby Charitable Foundation. Liam Paninski receives support from a NSF CAREER award.

## Footnotes

[1]Our task is control of the BCI for arbitrarily long duration, so it makes sense to look for the stationary decoder. Similarly the BCI user will look for a stationary encoder. We could also handle the finite horizon case (see section 2.3 for further discussion).

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
