[Supplementary Material]

**Supplementary Materials**

# 1  LQR

Here we derive the standard LQR solution and connect the general solution to the updates of the encoding model. Consider the dynamics

$$x_{t+1} = Ax_t + Bu_t + \epsilon_t; \quad \epsilon_t \sim \mathcal{N}(0, C)$$

where $u_t$ is a control signal that depends on $x_t$. We want to choose $u_t(x_t)$ to minimize the expected quadratic cost with rate

$$x_t^T Q x_t + u_t^T R u_t + 2x_t^T N u_t.$$

We can write down the optimal expected cost-to-go $v_t$ given $x_t$ (the Bellman equation). As we do this, we will see by backward induction that this cost-to-go is quadratic in $x_t$, so let $V_t$ be its Hessian.

$$v_t(x_t) = \min_{u_t}[x_t^T Q x_t + u_t^T R u_t + 2x_t^T N u_t + \mathbf{E}[v_{t+1}(x_{t+1})]]$$

$$= \min_{u_t}[x_t^T Q x_t + u_t^T R u_t + 2x_t^T N u_t + (Ax_t + Bu_t)^T V_{t+1}(Ax_t + Bu_t) + \operatorname{tr}(V_{t+1}C) + v_{t+1}(0)].$$

We differentiate by $u_t$ to find the optimal control

$$Ru_t + N^T x_t + B^T V_{t+1}(Ax_t + Bu_t) = 0$$

$$u_t = -(R + B^T V_{t+1}B)^{-1}(N^T + B^T V_{t+1}A)x_t,$$

so the "feedback gain" is

$$L_t = -(R + B^T V_{t+1}B)^{-1}(N^T + B^T V_{t+1}A),$$

and we substitute to get

$$v_t(x_t) = x_t^T Q x_t + x_t^T N u_t + (Ax_t)^T V_{t+1}(Ax_t + Bu_t) + \operatorname{tr}(V_{t+1}C) + v_{t+1}(0),$$

so

$$V_t = Q + A^T V_{t+1}A - (N + A^T V_{t+1}B)(R + B^T V_{t+1}B)^{-1}(N^T + B^T V_{t+1}A)$$

and

$$v_t(0) = \operatorname{tr}(V_{t+1}C) + v_{t+1}(0).$$

Now suppose that there's a variable $z_t$ which (given $u_t$) is jointly Gaussian with $x_t$

$$z_t = Fx_t + Gu_t + \xi_t; \quad \xi_t \sim \mathcal{N}(0, H),$$

and an additional cost term quadratic in $z_t$

$$v_t(x_t) = \min_{u_t} \mathbf{E}[x_t^T Q x_t + u_t^T R u_t + 2x_t^T N u_t + z_t^T S z_t + v_{t+1}(x_{t+1})]$$

$$= \min_{u_t}[x_t^T(Q + F^T SF)x_t + u_t^T(R + G^T SG)u_t + 2x_t^T(N + F^T SG)u_t + \operatorname{tr}(SH) + \mathbf{E}[v_{t+1}(x_{t+1})]],$$

which is like having

$$Q' = Q + F^T SF$$
$$R' = R + G^T SG$$
$$N' = N + F^T SG.$$

In the LQG setting, we would take $z_t$ to be the actual hidden state, and $x_t$ the Kalman estimate of this state. In the Kalman filter, $x_t$ and $z_t$ have the same dynamics except that $x_t$ is noisier, i.e. they have the same $A$ and $B$ but different $C$ (which we don't care about). Now $z_t$—$x_t$—$u_t$, so that $G = 0$, and the filter is designed to have $F = I$. $Q = 0$ because the cost is on the real state, so $Q' = S$, $R' = R$, $N' = N$. $H$ is the conditional covariance which is sometimes denoted $\Sigma$, but we don't care about it either (this where the separation of LQE and LQR comes from). This all means that LQR is the same whether applied to the actual hidden state or its Kalman estimate.

In our case, the control depends on $\begin{bmatrix} x_t \\ \hat{x}_{t-1} \end{bmatrix}$, while the cost depends on $\begin{bmatrix} x_t \\ \hat{x}_t \end{bmatrix}$. Also note that the $u_t$ in this section is before the neural noise $\eta_t$ is added. So here's how things in this supplement relate to things in the paper:

| This Supplement | The paper |
|---|---|
| $x_t$ | $\begin{bmatrix} x_t \\ \hat{x}_{t-1} \end{bmatrix}$ |
| $u_t$ | $Ax_t + B\hat{x}_{t-1}$ |
| $A$ | $\begin{bmatrix} P & 0 \\ 0 & G \end{bmatrix}$ |
| $B$ | $\begin{bmatrix} 0 \\ (FC) \end{bmatrix}$ |
| $C$ | $\begin{bmatrix} Q & 0 \\ 0 & (FC)R(FC)^T + FSF^T \end{bmatrix}$ |
| $Q$ | $0$ |
| $R$ | $\tilde{R}$ |
| $N$ | $0$ |
| $z_t$ | $\begin{bmatrix} x_t \\ \hat{x}_t \end{bmatrix}$ |
| $F$ | $\begin{bmatrix} I & 0 \\ 0 & G \end{bmatrix}$ |
| $G$ | $\begin{bmatrix} 0 \\ (FC) \end{bmatrix}$ |
| $H$ | $\begin{bmatrix} 0 & 0 \\ 0 & (FC)R(FC)^T + FSF^T \end{bmatrix}$ |
| $S$ | $\begin{bmatrix} I & -I \\ -I & I \end{bmatrix}$ |
| $Q'$ | $\begin{bmatrix} I & -G \\ -G^T & G^TG \end{bmatrix}$ |
| $R'$ | $\tilde{R} + (FC)^T(FC)$ |
| $N'$ | $\begin{bmatrix} -(FC) \\ G^T(FC) \end{bmatrix}$ |

## 2 Decoding improvement with perfect estimation

Supplementary figure 1 depicts the decoding performance on a sample realization of the data (for one initial condition of figure 4(a) in the main paper). Comparing the initial decoding performance with the final one, it is clear that this form of naive co-adaptive dynamics improve task performance.

|   |   |
|---|---|
| (a) Initial decoding performance | (b) Final decoding performance |

Figure 1: Each figure presents the 3 dimensions of $x_t$ (in blue) over simulated time (along the horizontal axis). Superimposed on each subplot is the decoded trajectories $\hat{x}_t$ (in red). (a) Decoding performance with a random initial encoder, and the decoder that is optimal for it. (b) Encoder-decoder pair has converged to an optimum joint setting and decoding quality is meaningfully improved.

# 3 RLS procedure

We propose a recursive least squares algorithm closely related to the Kalman filter (RLS-Kalman) as the method by which to estimate the parameters. This allows the estimates to change gradually as more data is generated, which in turn causes a gradual change in the parameters computed by the smooth update steps.

We first present a general form of the RLS-Kalman estimation method we use, and subsequently specialize it to the two cases where it is used symmetrically (by each agent to estimate the parameters of the other). We suppose that the vectorized matrix $M$ to be estimated has dynamics:

$$M_t = \lambda M_{t-1} + \omega_t; \quad \omega_t \sim \mathcal{N}(0, W),$$

where $W = \sqrt{1 - \lambda^2}I$ and $0 < 1 - \lambda < 1$ is a forgetting factor which allows for gradual change in the parameters. The observation equation is

$$z_t = H_t M_t + \nu_t; \quad \nu_t \sim \mathcal{N}(0, U).$$

The Kalman filter update allows for online estimation of the parameter matrix $M_t$ at each timestep as new inputs $H_t$ generate new observations $z_t$. The updates take the form

$$K_t = (\Sigma_{t-1} + W)H_t^T(H_t(\Sigma_{t-1} + W)H_t^T + U)^{-1}$$
$$\hat{M}_t = \lambda \hat{M}_{t-1} + K_t(w_t - H_t \lambda \hat{M}_{t-1})$$
$$\Sigma_t = (I - K_t H_t)(\Sigma_{t-1} + W).$$

For the decoder-side agent, estimation is of $M = vect\{[CA \quad CB]\}$ which corresponds to the partially observable encoding model. In this case $H_t = \begin{bmatrix} x_t \\ \hat{x}_{t-1} \end{bmatrix}^T \otimes I$ and the observation is $z_t = y_t$.

For the encoder-side agent, estimation is of $M = vect\{\begin{bmatrix} FC \\ G \end{bmatrix}\}$ which corresponds to the partially observable decoding model. In this case $H_t = \begin{bmatrix} u_t \\ \hat{x}_{t-1} \end{bmatrix}^T \otimes I$ and the observation is $z_t = \hat{x}_t$. For simulations, $M$ is initialized to zeros and its covariance is initialized to the $I$ – these match the prior from which the encoding model is drawn.

# 4   Simulation parameter selections

For the various simulations, reasonable selections of parameters will reflect the biological realities (1) that individual neurons are noisy, (2) that we observe a subset of the neurons, and (3) that for numbers of electrodes that we observe, neural noise isn't entirely averaged out. For our simulations, there is no absolute measure of scale for some of the values (such as the noise), so we first fixed the number of electrodes and tuned neural noise such that decoding was noisy, but such that there would be some signal as is experimentally observed. We treat electrode noise as negligible. Our code is available from the first two authors' research websites.

We simulate a population of 50 neurons which we responsive to our 3 degrees of freedom. We observe "electrode" signals from 5 neurons. In a real experiment with 10s of neurons recorded, only a subset of neurons will respond for any particular degree of freedom. Our relatively small number of electrodes should be considered comparable to the number of electrodes with fairly responsive signals for the purposes of decoding.

We tuned the neural signal cost for the user $\tilde{R}$ such that the neural signal could not increase arbitrarily to be $1e^{-2}$. For our choices of noise and number of electrodes, this parameter value sufficiently restricted the magnitude of the neural signal, however with other values for these other parameters, $\tilde{R}$ would need to be different.

For the simulations in which each agent estimates the parameters of the other (sections 4 & 5 of the main paper), there are some additional parameters. The penalty on the anticipated electrode signal changes $\tilde{R}'$ was larger than the actual neural $\tilde{R}$. This provided some regularization on the anticipated changes. This value was not finely tuned, and presumably the encoder aware updates could be further improved if this were chosen more thoroughly. Also, the $(1 - \lambda)$ values corresponding to the forgetting factors in the RLS procedure (supplement section 3) ranged between $1e^{-3}$ and $1e^{-6}$ with the first number corresponding to less memory and the latter number corresponding to almost perfect memory.