[Reviews · NeurIPS 2013]

Submitted by Assigned_Reviewer_3

Recently, there have been great interests to the co-adaptive Brain Computer Interface (BCI) to efficiently decode movement intentions. In this respect, this paper introduces a new framework that finds optimal parameters in encoder and decoder using a linear quadratic Gaussian system. The simulation results showed the parameter convergence in terms of a decoding error.

To optimize both the encoder and the decoder in BCI, the authors suggested the LQG-based multi-agent control method. However, this reviewer believes that it lacks the description of the characteristics of LQG problem and why this framework is proper to solve problems in real BCI condition.

The acquisition or composition of the simulated data was not clearly described, which makes it hard to understand the proposed method. Furthermore, please discuss how the proposed optimization rules can be successfully adapted to noisy and intrinsically non-stationary brain signals as a further research.

The authors claimed that the proposed method is better than standard co-adaptive approaches. Regarding the point, this reviewer believes that it would be beneficiary to provide additional information on the results such as trade-off comparison, etc.
Summary: This paper introduces a new framework that takes into account the changes made in the other side, in this paper, they call each side as an agent and formulates the co-adaption model using a linear quadratic Gaussian system. Despite a simulation-based description, the proposed system is well described and thus has a great potential for real applications.

Submitted by Assigned_Reviewer_4

In brain-computer interface (BCI), co-adaptation is an important issue, because not only the decoders, but also the brain activities can also adapt through feedback. To investigate this problem, the paper introduced a split linear quadratic Gaussian (LQG) system. Based on the theoretical framework used in the multi-agent control theory, the authors proposed interleaved algorithm combining linear-quadratic estimation (LQE) for decoder updates and linear-quadratic regulator (LQR) for encoder updates with recursive least squares (RLS) to estimate the system parameters of the other agents (Procedure 1). They modified further the co-adaptive decoder by incorporating r-step ahead estimators of the encoder parameters (Procedure 2). For proof of concepts, a numerical experiment with both procedures was conducted and the results looks promising.

Quality:
Although I did not check all the equations, the paper is technically sound except for the second line of Equation (13). It should be a scaler, but the matrix multiplications do not become scalars, probably. The authors employed a simple LQG system to investigate the co-adaptation problem. The simulation results nicely demonstrated that the procedures work as intended in a rather ideal situation. However, it is not clear from the manuscript how useful their algorithms are in actual BCI applications.

Clarity:
The paper is well organized and clear enough.

Originality:
I was not aware of a theoretical framework for investigating co-adaptation problem in BCI community (I have not read important references). The authors have in mind 3D tracking applications of BCI with invasive devices, where the LQG system is more appropriate. As future extensions, incorporating some nonlinearity is necessary to deal with other type of BCI applications.

Significance:
Although there is no validation of the proposed procedures with practical BCI applications, I think this work could inspire many BCI researchers who are looking for efficient feedback schemes to enhance BCI performances and to increase the number of BCI users.


Based on the above criteria, I would think that this paper is a fair contribution marginally above the threshold. One reason is that such a theoretical framework could inspire other researchers in the field.
Summary: Although this paper does not contain any real examples to show its practical importance and the LQG system may be too simple to deal with many other BCI applications, I think it is still nice for BCI community and relevant research areas that this work will be presented at NIPS. However, the manuscript is not so strong because of no practical validation. Thus, I would say that it is marginally above the threshold.

Submitted by Assigned_Reviewer_6

The authors attempted to develop a general theoretical framework for designing the co-adaptation between the encoder model and decoder algorithm in a brain-computer interface (BCI). Because the nervous system change its encoding properties in presence of sensory feedback, it has long been hypothesized that introducing co-adaptation between encoder and decoder will improve the performance of BCI. Perhaps the biggest objection to this hypothesis is that the co-adaptation may result in crazy oscillation in the parameters of the encoder model and the decoder, and lead to unstable BCI performance.

Through computer simulations with idealized parameters, the authors showed that the parameters of the encoder model converge to a stablized level, and co-adaptation led to lower estimation errors. Overall, this is an interesting theoretical exploration for the BCI community.

However, the authors made a lot of assumptions that may or may not be consistent with experimental knowledge. For example, the choices for the signal noise covariance Rc, the cost values, and the forgetting factor lambda were all explained vaguely, it's not clear whether they reflect biological reality. Also the comparison with other co-adaptation methods is lacking.

Overall, the technical quality is sufficient. The presentation is clear. Although there are certain drawbacks, it still represents an very informative exploration which will serve as a baseline to help inspire more novel ideas, and the results are sufficiently significant for publication in NIPS.
Summary: It would be great if the authors can address the concerns raised above. As is, the merits of this paper still warrant its publishing in NIPS.
Author Feedback

Author rebuttal: We appreciate all of the reviewers' comments. We address the points of all of the reviewers together.

In contemporary BCI applications, Kalman Filter (KF) based methods are widely used, relying on the same modeling assumptions used in this paper. These models were demonstrated to be adequate approximations, and can be expected to be equally applicable in our context. Some works use alternative methods, however their modeling choices are still under evaluation (e.g. which nonlinearities to use). It is natural then to introduce our framework in the LQG setting, validated by the ubiquity of KF-based methods in application, and extend to nonlinear settings in future work.

Previous works in co-adaptation characterize changes in performance when a user learns how to control a BCI in the presence of an adaptive decoder. In previous work, little emphasis has been placed on modeling how the full system, including the user, changes. We believe that introducing our modeling scheme in which both user and decoder adaptation are modeled is one of our contributions, and it isn't clear how one should compare its performance to that of existing approaches.

Our parameter selection was inspired by a biologically realistic parameter regime, as outlined in the end of section 3, although it wasn't fit to values from a specific experiment. Based on reviewer feedback, we intend to supplement the paper with details of the parameters used in our simulations, as well as publicly share our code. Our simulations were fairly robust, and the specific choices of parameters are incidental to the paper's contribution.

We appreciate the reviewers' recognition that our work may serve to motivate targeted experiments. While the focus in this paper is on simulations, we believe our approach to be of interest to the BCI experimental community, and hope that experiments will inspire further theoretical insights which we can build on. Our motivation in sharing our work at this stage is to provide novel theoretical tools to the community, while also driving the advancement of theoretical approaches. In future efforts, we will seek to extend the theoretical tools and begin applying them to experiments.

Again, thank you for your feedback.

Equation correction:
Equation (13) should be written with the vector (G\hat(x)_{t-1} - x_t) instead of the terms [-x_t G\hat(x)_{t-1}].